# Women Reduce the Performance Difference to Men with Increasing Age in Ultra-Marathon Running

**DOI:** 10.3390/ijerph16132377

**Published:** 2019-07-04

**Authors:** Karin J. Waldvogel, Pantelis T. Nikolaidis, Stefania Di Gangi, Thomas Rosemann, Beat Knechtle

**Affiliations:** 1Institute of Primary Care, University of Zurich, 8091 Zurich, Switzerland; 2Exercise Physiology Laboratory, 18450 Nikaia, Greece; 3School of Health and Caring Sciences, University of West Attica, 12243 Athens, Greece; 4Medbase St. Gallen Am Vadianplatz, 9001 St. Gallen, Switzerland

**Keywords:** age of peak performance, athlete, sex difference, ultra-endurance

## Abstract

Age and sex are well-known factors influencing ultra-marathon race performance. The fact that women in older age groups are able to achieve a similar performance as men has been documented in swimming. In ultra-marathon running, knowledge is still limited. The aim of this study was to analyze sex-specific performance in ultra-marathon running according to age and distance. All ultra-marathon races documented in the online database of the German Society for Ultra-Marathon from 1964 to 2017 for 50-mile races (i.e., 231,980 records from 91,665 finishers) and from 1953 to 2017 for 100-mile races (i.e., 107,445 records from 39,870 finishers) were analyzed. In 50-mile races, race times were 11.74 ± 1.95 h for men and 12.31 ± 1.69 h for women. In 100-mile races, race times were 26.6 ± 3.49 h for men and 27.47 ± 3.6 h for women. The sex differences decreased with older age and were smaller in 100-mile (4.41%) than in 50-mile races (9.13%). The overall age of peak performance was 33 years for both distances. In summary, women reduced the performance difference to men with advancing age, the relative difference being smaller in 100-mile compared to 50-mile races. These findings might aid coaches and ultra-marathon runners set long-term training goals considering their sex and age.

## 1. Introduction

The oldest entry in the collection of ultra-marathon running statistics provided by the “German Society for Ultra-Marathon” [1] was a 89 km run from London to Brighton taking place in 1837. Since then, the popularity of ultra-marathon running has substantially increased [2,3,4,5]. Ultra-marathon running competitions are mainly specified by duration in hours or days (e.g., six hours to ten days) or by distance in km or miles (e.g., 50 km, 100 km, 50 miles, and 100 miles). For a race to be considered as an ultra-marathon, the duration has to be at least 6 hours, or the distance has to be longer than 42.195 km (26.2 miles) [5,6].

Over the last decades, the number of ultra-marathon competitions [7] as well as the number of participants in these races has increased exponentially [8]. This increase appears to be mostly due to increasing numbers of athletes aged over 40 years (i.e., master athletes) [7], as well as women increasingly participating [3,8]. While very few women participated in the first ultra-marathon running competitions, their share has increased ever since [7,9,10]. Since 2004, approximately 20% of the runners have been women, but there are no records documenting women participating in the USA 161 km ultra-marathon distance in the 1970s [7].

Multiple determinants of the ultra-marathon’s success have been identified. One very important factor is age [10,11,12]. Knowing the age of peak performance has been assessed as being indispensable for optimization of the training schedule and to plan a successful career as an ultra-runner [13]. Comparing marathon and ultra-marathon running, differences in the age of peak performance have been very recently reported. In marathon running, the best performances of women and men are achieved between 25 and 35 years of age [4,11,14,15,16]. In above-marathon distances, the age of peak performance is higher [4]. Several studies report that the best results are observed in men aged 30 to 49 years and in women aged 30 to 54 years in 100 km ultra-marathon races [13,17]. One explanation might be that most runners start their careers with marathons, only later in life enhancing the challenge with ultra-marathons [17]. Moreover, compared to marathon races, ultra-marathons require the necessary level of performance, and this depends even more on the critical factors of adequate training preparation with an appropriate nutrition plan and mental strength [6].

Another essential factor influencing race performance is the athlete’s sex. Even though the performance of women compared to men in endurance running was inferior in the past [9], the sex-related gap has decreased in the last couple of decades [18]. This observation led to speculation about whether and how women could reduce the difference in running times to a level where they might outperform men in long-distance races. Other authors have hypothesized that this might be more likely to happen with very long distances, such as in ultra-marathons [18,19,20,21]. In contrast to such expectations, some results seem to indicate a larger sex gap in ultra-marathons compared to marathons [20], although there might be a potential bias underlying these results. Most studies either did not consider all participants and only focused on the top athletes [15,22,23], or had a limited sample size of athletes, only investigating a small number of races and/or a limited period [11,16]. Comparing only the top ten world record performances carries the risk of the results being affected by athletes with the highest performance level. For example, Lepers et al. [22] restricted their analysis of triathletes to the top ten men of each age group in the Olympic triathlon and Ironman triathlon world championships of 2006 and 2007 (440 athletes in total) and found an age-related performance decline at the age of 50 years in swimming and at the age of 45 years in cycling and running. In contrast, Käch et al. [24], investigating 329,066 men and 81,815 women participating in Ironman triathlon competitions held between 2002 and 2015, found a performance decline at profoundly earlier ages (in swimming, at 25–29 years of age in women and men; in cycling and running, at 30–34 years of age in women and at 35–39 years of age in men). According to Käch et al. [24], the participants in Ironman triathlons are not only the top-performing athletes of each age group, but also recreational athletes, the latter typically not being included in an analysis of top-ten athletes. Top-performing athletes tend to have more experience, mental strength, training volume, and training intensity than recreational athletes and are thus more likely to be included in analyses of top-ten performers [24]. This could also explain that the performance of unselected athletes (i.e., investigation of performance of every participating athlete, as found by Käch et al. [24], tends to decline at an earlier age than performance in top-ten athletes (as found by Lepers et al. [22]).

Could a similar mechanism also explain discrepant findings on the performance of men compared to women? Recent studies investigating master swimmers in pool and open-water swimming showed that women in older age groups (80 years and older) achieved a similar performance to men in an investigation of 65,584 freestyle pool swimmers (29,467 women and 36,117 men) competing in 50 to 800 m [25] races and when 7592 freestyle open-water swimmers (2829 women and 4768 men) competing in 3000 m [26] races in the FINA World championships from 1986–2014 and 1992–2014, respectively. In contrast, Senefeld et al. [27], who conducted a similar study except that they focused on the top ten swimmers in the years between 1986 and 2011 (6760 athletes in total, men and women), found that the performance of women in every age group was inferior and, contrary to Knechtle et al. [25,26], that the sex gap increased with age. 

The differences in the selection of performance levels could possibly explain these discrepant findings in comparisons of men versus women. In support of this interpretation, other studies found a reduction in the sex gap in swimming performance with increasing age for different disciplines such as breaststroke [28], backstroke [29], butterfly [30], and in the individual medley event [31]. The commonality across these studies is that they included all participants in their investigation, rather than only the top ten performance participants. The results indicate that selection based on performance levels has an influence on the results regarding sex-specific performance differences, usually in favor of men. In contrast, studies performed with all athletes tend to display a lower or no sex gap. 

The fact that women in older age groups (i.e., older than 80 years) achieve a similar performance to men has only been reported for different swimming disciplines, but not for running. Knechtle et al. [32] reported results on ultra-marathon performance in men and women and found that with increasing age and race distance, the sex gap increased rather than decreased. However, our knowledge of whether women in older age groups would be able to achieve a similar performance for longer running distances is still limited. For instance, a recent study on road running records from 5 km to 6 days showed that men were faster than women, the sex gap decreased with increasing age, and it did not vary by race distance or duration [33]. 

We therefore investigated whether women in 50-mile and 100-mile ultra-marathon races would be able to reduce the gap to men in older age groups. In contrast to previous studies, we analyzed a much larger data set, containing all 50-mile ultra-marathon races held between 1964 and 2017 and all 100-mile races between 1953 and 2017, thus avoiding selection bias by not only focusing on the top participants. Based on previous findings for master swimmers, we hypothesized that the sex gap in performance in ultra-marathons would decrease with increasing age, and that this decrease would be independent from the race distance.

## 2. Materials and Methods 

### 2.1. Ethical Approval

This study was approved by the Institutional Review Board of the Kanton St. Gallen, Switzerland, with a waiver of the requirement for informed consent of the participants, as the study involved the analysis of publicly available data (1 June 2010).

### 2.2. Data Sampling

The investigation comprised all ultra-marathon competitions with running distances documented in “miles” in the online database of the German Society for Ultra-Marathon (Deutsche Ultramarathon Vereinigung e.V.). A total of 7769 competitions with 456,167 men and women participating in the years from 1928 to 2017 [34] were extracted.

The data set was retrieved in multiple steps. First, we used the Google Chrome browser (Version 66.0.3359.139) with the add-on “Web Scraper” (Version 0.3.7) to retrieve the Uniform Resource Locator (URL) of each ultra-marathon competition registered in the online database. Each URL was saved in Microsoft Excel 2013 (Version 15.0.4569.1504). Subsequently, the Microsoft Excel-integrated Visual Basic Application (VBA) was used to filter the database contents, excluding every URL of competitions that did not have a distance specified in miles. In a final step, also using Excel-VBA, the raw data of each competition was extracted and uniformly formatted. The resulting file was visually controlled for inconsistences, and these were corrected in accordance with the original data. For the purpose of the present study, we analyzed 339,425 records of athletes either finishing a 50-mile race from 1964 to 2017 or a 100-mile race from 1953 to 2017. Other race distances were excluded due to insufficient data. The following variables were extracted: year of race, race distance, name of race, country of race, race time (h), running speed (km/h), name of athlete, year of birth, nationality of athlete, and sex of athlete. Age was derived by subtracting the year of birth from 2017.

### 2.3. Statistical Analysis

Descriptive statistics are presented as means ± standard deviations. Performance, or race time, was recorded in the format “hours:minutes:seconds” (h:min:s) and converted into hours, as a numerical variable. For 50- and 100-mile ultra-marathon races, t-tests were performed to compare the average performance between men and women by age group and by country. It was acknowledged that analyses of variance (ANOVAs) might have been easier to interpret; however, the mixed regression analysis was preferred since it was necessary to correct for clustered observations within runners who participate more than once. ANOVA would have not accounted for clustered observations. The age groups were 10–19, 20–29, 30–39, 40–49, 50–59, 60–74, and 75–95 years, and only observations with non-missing ages were considered in analyses involving age. Country groups were identified through participation prevalence by country: United States of America (USA), Canada (CAN), Great Britain (GBR), and Republic of South Africa (RSA). The other countries were grouped together. Age was considered as a continuous variable, in 1-year intervals, when defined as a predictor variable for ultra-marathon time. A non-linear regression mixed model with basis splines was performed to find the age of peak performance, which is the age at which the time record-fitted value has a minimum. The mixed model was used to correct for repeated measurements within runners (clusters) through the random effects of intercepts. Different regression model specifications were initially considered, with age–sex, age–country, and country–sex interaction terms and with different hypotheses about the age and time trend. Model selection was performed using both the Akaike information criterion (AIC) and the Bayes information criterion (BIC). In the final selected model, age, calendar year, sex, country, and a country–sex interaction term were considered as fixed effect predictors. The statistical model was specified as follows: Ultra−marathon time (Y)~ [fixed effects (X) = BS(year, df=3) + BS(age, df=3) + sex + country + country∗sex] + [random effects of intercept=runners] where BS (year, df=3) and BS (age, df = 3) are 3 degrees of freedom (df) basis splines changing with calendar year and age, respectively; country*sex denotes the country–sex interaction term.

Two different analyses were performed, one for 50-mile and one for 100-mile races. In the 50 miles analysis, South Africa was combined with other countries because of the low number of runners. Results of the regression models are presented as estimates and standard errors. In addition, sex differences (%) in performance were examined, defined as 100 × (women’s race time-men’s race time)/men’s race time. For all tests and regressions, statistical significance was defined as *p* < 0.05. All statistical analyses were carried out with R [35]. The packages ggplot2, lme4, and lmerTest were used, respectively, for data visualization and for the mixed model. 

## 3. Results

Between 1964 and 2017, a total of *n* = 231,980 records on 91,665 different finishers with information on age were retrieved from the database on 50-mile ultra-marathon races. For 100-mile races, a total of *n* = 107,445 records on 39,870 different finishers was available for the period between 1953 and 2017. Overall, the average number of observations per runner was 2.53 in 50-mile and 2.69 in 100-mile races. In 50-/100-mile races, the number of women was 23,548 (26%)/7789 (20%) with 55,540 (24% of the total observations)/20,154 (19% of the total observations) records, and the number of men was 68,107 (74%)/32,081 (80%) with 176,440 (76%)/87,291 (81%) records. 

The proportions of observations of finishers aged 50 years and above were 24.4% (men) and 17% (women) in 50-mile races and 24.9% (men) and 18.3% (women) in 100-mile races, indicating that finishing men tended to be slightly older than women. The vast majority of finishers participated in races in the USA (85.2%); 6.1%, 3.8%, and 0.1% of the sample participated in Great Britain, Canada, and South Africa, respectively, and 4.1% in races taking place in 43 other countries. 

In Table 1, the number of observations and the average performance by sex, age group, and country are reported for 50- and 100-mile races. In both 50-/100-mile races, the shortest average race times were observed in the 20–29 years age group, both in men (10.30 h/26.07 h) and in women (11.18 h/27.14 h); the lowest average performances were observed in the 75–95 years age group, again both in men (14.20 h/29.73 h) and in women (13.40 h/29.00 h). In 50-mile races, the shortest average race times were observed in Canada (10.47 h in men and 11.35 h in women) and the longest in Great Britain (11.67 h in men and 12.87 h in women). In 100-mile races, the shortest average running times occurred in South Africa (21.82 h in men, 23.19 h in women) and the longest in the group of the 43 “other” countries (27.73 h in men and 28.11 h in women). Performance differences between sexes were significant (*p* < 0.001) for all age groups <75 years in the 50-mile races and for age groups between 20 and 60 years in the 100-mile races. In the ≥75 years of age group, better performances occurred in women compared to men, even though the difference failed to attain statistical significance due to the small sample size. The magnitude of the difference was, however, similar to that seen in younger age groups, where men are faster than women, and particularly in 5- mile races. With the largest performance sex gap in favor of men seen in the youngest age group (10–19 years), a clear performance trend over age is visible for both distances.

Regarding country, in both distances and for all country groups, performance differences were significant (*p* < 0.001) between sexes due to a better performances in men, the largest differences being observed in South Africa. Table 2 describes, for both distances, the results of the statistical models, as described in the methods section (model selection statistics omitted). For 50 miles, race times were 11.74 (Sd = 1.95) h for men and 12.31 (Sd = 1.69) for women, with a sex difference of 9.13%. For 100 miles, race times were 26.6 (Sd = 3.49) h for men and 27.47 (Sd = 3.6) h for women, with a sex difference of 4.41%. Women were significantly slower than men (*p* < 0.001), the estimated sex differences being 0.74 (SE = 0.017) and 0.81 (SE = 0.075) hours in 50- and 100-mile races, respectively. For 50 miles, compared to the USA, finishers in Canada and in other countries were significantly (*p* < 0.001) faster by 0.19 (SE = 0.040) and 0.092 (SE = 0.028) hours, respectively. In contrast, finishers in GBR were significantly (*p* < 0.001) slower by 0.656 hours. For 100 miles, compared to the USA, finishers in GBR, CAN, and the RSA were significantly (*p* < 0.001) faster, with runners in the RSA being faster than in the USA by an estimated 3.938 hours. Other countries were slower by 0.893 hours, *p* < 0.001, compared to the USA.

The Country*Sex interaction terms, for example the term Great Britain*W, estimates how much greater the effect of being a woman in a particular country (e.g., Great Britain) was on race time, compared to the USA. The interaction effects (Table 2) are visualized in Figure 1 (50 miles) and Figure 2 (100 miles). They were particularly pronounced for GBR in 50-mile races (0.562 hours) and for CAN in 100-mile races (0.751 hours), where the performances of women and men differed clearly more than in the USA. The distance between the fitted curves in men and women is largest for GBR in 50-mile races and for CAN in 100-mile races.

All the effects in Table 2, together with the age and year of peak performance, are shown graphically in Figure 1, Figure 2, Figure 3 and Figure 4. Both in 50-mile (Figure 1) and 100-mile (Figure 2) races, running times decreased and, after reaching a minimum at 33 years (peak performance), increased with increasing age. Regarding the calendar period, in 50-mile races, 1985 was the year of the best performance (Figure 3), whereas in 100-mile races, performance worsened consistently over time (Figure 4). In Figure 5, the estimated sex differences in performance by country are shown over age. For both distances, the differences in favor of men increased up to about 33 years, and the increase was subsequently followed by a decrease. In 100- but not in 50-mile races, the differences re-increased slightly after about 80 years of age. For both distances, the estimated sex differences were smaller for 100- than for 50-mile races. Over calendar time, from 1953 to 2017, the sex difference in performance decreased continuously in all countries in 100-mile races. For roughly the same period, the sex difference in performance peaked at around 1985 in all countries for 50-mile races (Figure 6).

## 4. Discussion

The aim of this study was to examine the sex gap in performance in ultra-marathons. We hypothesized a decrease of the sex gap with increasing age and that this decrease would be independent from race distance. The main findings were that the (i) sex difference in performance was smaller in older than in younger athletes; (ii) the relative sex difference in performance was smaller in 100- than in 50-mile races; (iii) the sex difference in performance approaches a historical minimum; (iv) the peak performance age was 33 years; (v) the average performance worsened over the last three decades. Minor findings were that (vi) men were slightly older than women; (vii) more than two thirds (70%) of the finishers had participated in 50-mile races; (viii) three quarters (76%) of all finishers were men; (ix) the proportion of men was higher in 100-mile races (80%) than in 50-mile races (74%); (x) in South African races, men and women demonstrated the best 100-mile performances. 

### 4.1. The Sex Difference in Performance Was Smaller in Older Than in Younger Athletes

In 50-mile races, the decline in the sex difference always decreasing up to the highest age. There are multiple possible physiological mechanisms in men for the reduction in the performance sex gap with increasing age, including lower levels of anabolic hormones [37], a decrease in neuromuscular efficiency [38], and a reduced ability to synthesize protein [39] as well as body fat [40]. In addition, the loss in skeletal muscle mass is more pronounced in men at the age of 60 years and above compared to women of the same age, with sarcopenia present in ~53% of men compared to ~47% of women [41]. Our finding of a sex gap reduction with increasing age in ultra-marathon running is consistent with recent findings of studies analyzing master swimmers competing in pool and open-water races [6,25,26,29,30,31]. The factor of sarcopenia was also suggested by Knechtle et al. [25], who investigated 65,584 freestyle master swimmers between 1986 and 2014. Sarcopenia might thus be an important factor in ultra-marathon running as well. Finally, compared to men, women tend to live longer and to be in better physical condition later in life [30]. A larger higher-age population of high-performing women as compared to men in 50-mile ultra-marathon races can thus be expected based on these considerations. 

In contrast to 50-mile races, in 100-mile races, the age-related downward trend in the sex difference reversed, the sex difference again increasing after about 80 years of age. It has to be noted, however, that the number of athletes in the oldest age group was rather small, in particular in 100-mile races. Thus, the increase in the sex gap in 100-mile races could simply be due to chance. Alternatively, however, the possibility of an increasing out-selection of relatively slow men at higher ages, in particular in 100-mile races, cannot be excluded. This does not appear completely implausible as physical performance is predictive of longevity at older ages [42,43], possibly underlying a deficit in high-performing men. However, as the increase of the sex gap at very high ages did not occur in 50-mile races, plain chance appears to be the more plausible explanation. 

### 4.2. Relative Sex Difference in Performance Was Smaller in 100- Than in 50-Mile Races 

Our second hypothesis of the decrease in the sex gap in performance with increasing age being independent from race distance was not confirmed. For both race distances, a decline of the relative sex difference is clearly visible, and the decline is more pronounced in 50-mile races than 100-mile races. One explanation for this finding could be that in extremely long distances, like 100-mile races, there might exist a sex-independent pace limit [44].

This limit might constitute a performance maximum, outweighing sex differences at increasing ages (“ceiling effect”). Nikolaidis et al. [11] also found a decrease in the sex gap with increasing race distance from half-marathon to marathon and to 100-km ultra-marathon races. In contrast, Coast et al. [20] found, more than 10 years earlier, an increasing sex gap with increasing running distances. However, these authors had restricted their analysis to world-best running performances at distances from 100 m to 200 km, and results might thereby have been biased by the selection of mostly top athletes. Furthermore, the authors indicate that their results might have been confounded by the reduced number of women in longer-distance events. The question of whether the sex gap depends on race distance thus requires further research.

### 4.3. Sex Difference in Performance Approaches a Historical Minimum

Over the period of 1953 to 2017 (end of study period), the sex difference decreased across all countries in 100-mile races. In contrast, in 50-mile races, the decline was restricted, across all countries, to the period of about 1985 to 2017, whereas in the preceding period from 1964 on, the sex difference had increased across all countries. For either distance, the range of sex differences between countries exceeds that within countries and is larger for 50- than for 100-mile races, the ranking of the countries being different in 50- and 100-mile races.

### 4.4. Peak Performance Age Was 33 Years

In spite of some outliers at certain ages and in certain countries, based on very few or even single individuals per year of age, as demonstrated in Figure 1 and Figure 2, the model-based peak performance age was 33 years, for both race distances. It has to be noted that due to the age*sex and age*country interaction terms being dismissed during the course of the pre-specified stepwise model-building process, the performance peaks do not show variation across sex and country. The model-based estimate of the age of peak performance of 33 years is in line with the available literature. Most previous studies suggested that the age of peak performance in ultra-marathons lies between 30 to 49 years for men and between 30 to 54 years for women [3,4,13,17,45]. However, our result is more in the suggested range of marathon peak performance age of 25 to 35 years for men and women [11,14,15,16]. 

### 4.5. Average Performance Worsened Over the Last Three Decades

In 50-mile races, the average running speed improved in both sexes from 1964 (start of the study period) up to 1985, subsequently worsening until the end of the study period (2017). In 100-mile races, a decline of the average running speed occurred over the whole study period, i.e., from 1953 to 2017. The performance improvement in 50-mile races up to 1985 is essentially attributable to US races, with relatively low average running speeds (see Figure 3). As about 85% of the races in the data set were from the USA, these particular races have inevitably impacted the overall shape of the performance-over-calendar-year-curve, the model specification not allowing for a country-specific shape due to not considering a country*calendar year interaction term. Had these low-performance US races been omitted from fitting the specified model, this would have resulted in a continuous performance decrease over the whole study period for 50-mile races, as it did for 100-mile races. Historically, the USA was the first country where ultra-marathon running became a popular activity among 436 recreational athletes [46]. It is reasonable to assume that initially, these recreational athletes had engaged preferably in 50-mile rather than 100-mile races, lowering the average performance. For example, 494,414 runners participated in 50-km ultra-marathon races between 1975 and 2016 [4], as compared to only 370,051 runners who participated in 100-km ultra-marathon races between 1959 and 2016 [13]. This is confirmed with the present data, where more than two thirds (70%) of the finishers had participated in 50-mile races. 

Despite this impact of the US-specific phenomenon in shaping the performance model curve, the general trend visible in the data is a performance decline over the study period. One factor for this general decrease in running speed across calendar years could be the popularity of ultra-marathon races gradually increasing worldwide and the races increasingly attracting recreational (i.e., master) athletes. As a consequence, the average performance would have gradually shifted to lower levels. It can be assumed that 100-mile races have always been less attractive to recreational athletes, as they require a more rigorous preparation than 50-mile races. An example for the more rigorous preparation with increasing race distances is provided by Rüst et al., who compared training characteristics between marathoners and 100-km ultra-marathoners and found that ultra-marathoners completed significantly more hours and kilometers during their training. It is therefore likely that the influx of recreational athletes into 100-mile races has been more gradual than in, for example, 50-mile races, and this trend leads to a non-linearity of the performance trend curve.

### 4.6. Limitations and Strength

A limitation of the present study was that it considered specific ultra-running race distances (50 and 100 miles) and thus, caution would be needed to generalize the findings to ultra-running races of other distances or durations [33]. On the other hand, a strength was the large data set that was available for analysis, which was not restricted to top athletes and covered the whole range of performance levels. The depth of the dataset both temporally as well as geographically, and the number of race distances included resulted in the opportunity to provide a comprehensive historic coverage of both 50- and100-mile ultra-marathon results. The data was publicly available, and the collection of data was independent from the data analysis, and therefore the replication of the present analyses is possible. The study findings may aid coaches and ultra-marathon runners in setting long-term training goals based on an athlete’s age and sex. For example, knowledge of the peak age of performance (33 years in both race distances) may influence individuals seeking to race in these distances and recruitment from a coaching perspective. Furthermore, the variation in performance by sex might also influence the training stimulus to be more homogeneous because the sex difference is small (e.g., 100 miles or elder age groups). In addition to performance, the abovementioned practical applications were also relevant from a health perspective. The role of exercise in the prevention and treatment of diseases (e.g., coronary artery disease, stroke, hypertension, diabetes, arthritis, osteoporosis, dyslipidemia, obesity, depression, cancer, and chronic obstructive pulmonary disease) has been well recognized [47]. The findings of the present study can aid physicians prescribing endurance exercise considering sex and age [48].

## 5. Conclusions

In summary, as age increases, the performance difference between women and men decreases and also becomes lower with longer distances. Based on the model, the overall age of peak performance was 33 years. Future investigations should not just include races measured in miles but also those measured in kilometers, and also analyze time-limited races. Potentially relevant covariates should be considered and whenever possible, acquired prospectively through interviews when they cannot be accessed through administrative databases. If this data had been available, a more comprehensive analysis could have been conducted, including covariate-adjusted time-series analysis.

## Figures and Tables

**Figure 1 ijerph-16-02377-f001:**
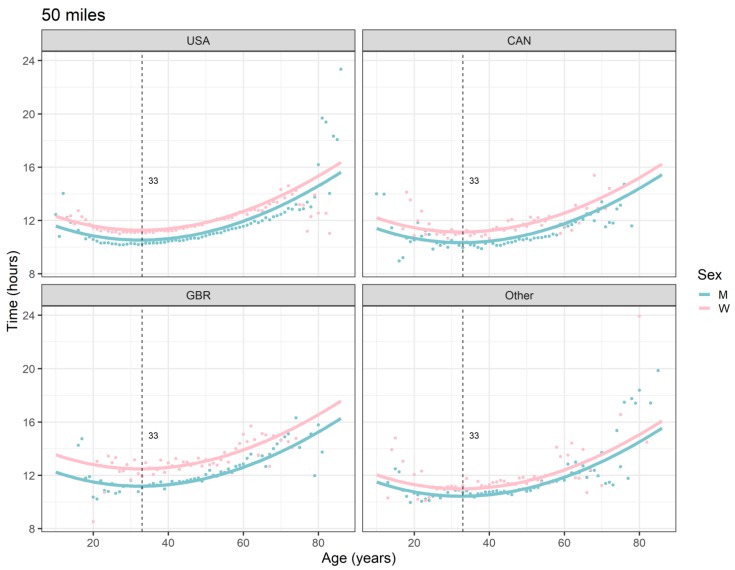
Ultra-marathon speed, 50 miles, by sex, age (in years), and country. Points are race-time averages. Lines are fitted curves (mixed model). Vertical lines with numeric labels are the ages at peak performance. USA = United States of America, CAN = Canada, GBR = Great Britain, W = women, M = men.

**Figure 2 ijerph-16-02377-f002:**
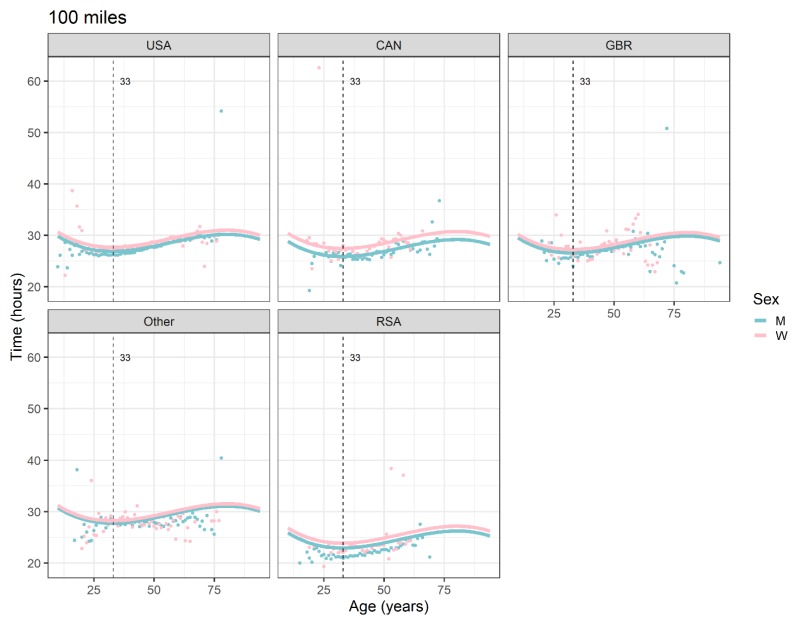
Ultra-marathon speed, 100 miles, by sex, age (in years), and country. Points are race-time averages. Lines are fitted curves (mixed model). Vertical lines with numeric labels are the ages at peak performance. The sample sizes decrease towards the minimum and maximum of the age axes, with some of the points reflecting only individuals; for example, the five points corresponding with GBR men 75+ years of age reflect one individual each, one of the three remarkable individuals (Geoffrey Oliver) accounting for three of the five points [36]. USA = United States of America, CAN = Canada, GBR = Great Britain, RSA = Republic of South Africa, W = women, M = men.

**Figure 3 ijerph-16-02377-f003:**
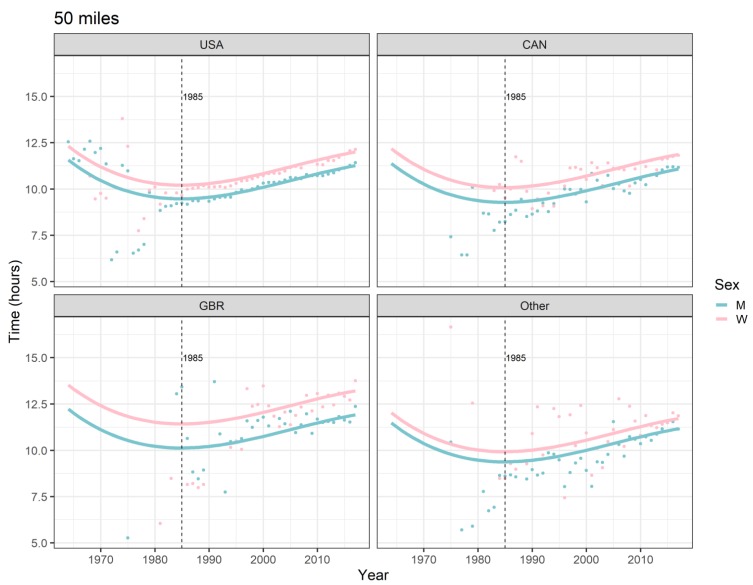
Ultra-marathon speed, 50 miles, by sex, calendar year, and country. Points are race-time averages. Lines are fitted curves (mixed model). Vertical lines with numeric labels are the ages at peak performance. USA = United States of America, CAN = Canada, GBR = Great Britain, W = women, M = men.

**Figure 4 ijerph-16-02377-f004:**
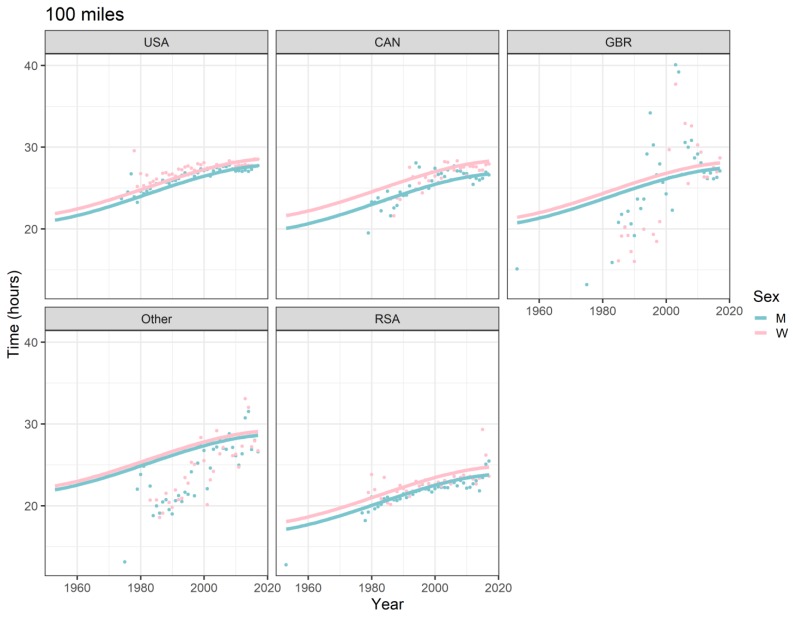
Ultra-marathon speed, 100 miles, by sex, calendar year, and country. Points are race-time averages. Lines are fitted curves (mixed model). Vertical lines with numeric labels are the ages at peak performance. USA = United States of America, CAN = Canada, GBR = Great Britain, RSA = Republic of South Africa, W = women, M = men.

**Figure 5 ijerph-16-02377-f005:**
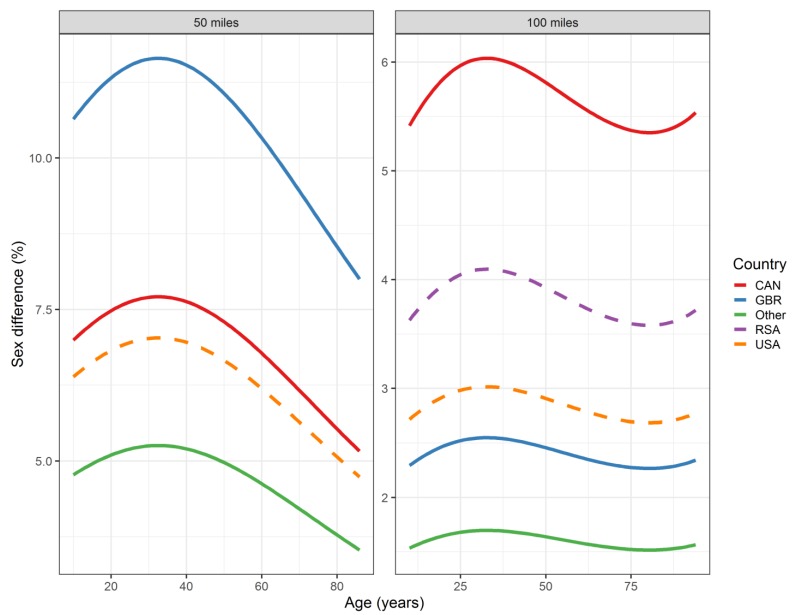
Sex differences by age (in years) and country in 50- and 100-mile ultra-marathons. Curves represent fitted values. For 50-mile races, South Africa was combined with other countries. USA = United States of America, CAN = Canada, GBR = Great Britain, RSA = Republic of South Africa, W = women, M = men. Sex differences (%) in performance were defined as 100× (women’s race time–men’s race time)/(men’s race time).

**Figure 6 ijerph-16-02377-f006:**
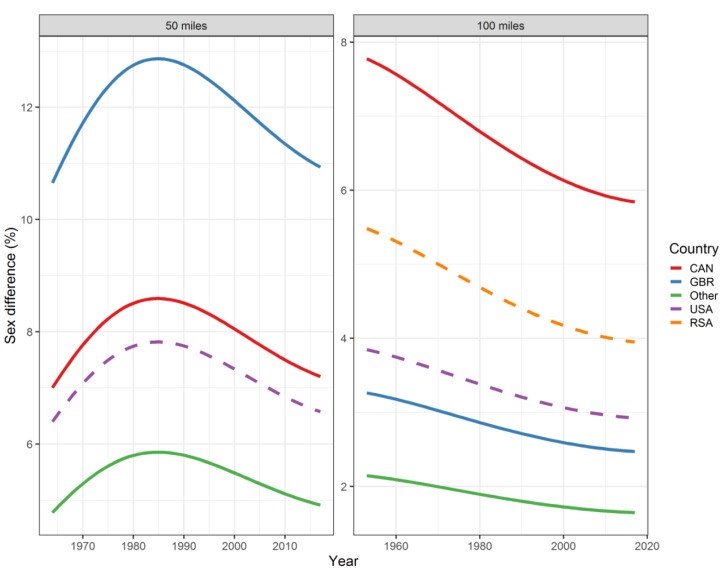
Sex differences by calendar year and country in 50-mile and 100-mile ultra-marathons. Curves represent fitted values. For 50-mile races, South Africa was combined with other countries. USA = United States of America, CAN = Canada, GBR = Great Britain, RSA = Republic of South Africa, W = women, M = men. Sex differences (%) in performance were defined as 100 × (women’s race time–men’s race time)/men’s race time.

**Table 1 ijerph-16-02377-t001:** Mean ultra-marathon performance (50 and 100 miles) by sex, age group, and country (South Africa, due to a small sample size for 50-mile races, is combined with other countries). *p*-values of a t-test of mean performance between sexes are shown.

		**50 miles, *n* = 231,980**				**100 miles, *n* = 107,445**			
**Age group**	**Sex**	***n***	**Mean (hours)**	**Sd (hours)**	***p***	***n***	**Mean (hours)**	**Sd (hours)**	***p***
10–19	Men	1312	10.9778	2.3152	<0.001	131	26.6158	5.1444	0.057
	Women	177	12.0706	2.0483		12	30.9698	7.0097	
20–29	Men	18,124	10.3022	2.1982	<0.001	5966	26.0697	5.4017	<0.001
	Women	6409	11.1797	2.2465		1410	27.1409	4.9447	
30–39	Men	53,553	10.3663	2.1998	<0.001	26,069	26.1852	5.6369	<0.001
	Women	19,256	11.2043	2.2210		6778	27.3619	5.1514	
40–49	Men	60,351	10.6421	2.1462	<0.001	33,387	26.9095	5.5481	<0.001
	Women	20,234	11.5077	2.2047		8257	27.7625	5.0640	
50–59	Men	33,857	11.1670	2.0841	<0.001	17,867	27.8913	5.2604	<0.001
	Women	8210	12.1018	2.1911		3350	28.5333	5.0671	
60–74	Men	9054	12.0289	2.1031	<0.001	3844	28.9254	4.9203	0.358
	Women	1230	12.9829	2.4215		342	28.6806	4.6994	
75–95	Men	189	14.1952	3.6604	0.199	27	29.7292	6.2894	0.571
	Women	24	13.4018	2.6675		5	29.0034	0.8413	
**Country**	**Sex**	***n***	**Mean (hours)**	**Sd (hours)**	***p***	***n***	**Mean (hours)**	**Sd (hours)**	***p***
Canada	Men	6208	10.4748	2.2036	<0.001	2924	26.2718	4.4521	<0.001
	Women	2563	11.3481	2.2938		1000	27.3393	4.6320	
Great Britain	Men	11,249	11.6678	3.0373	<0.001	4724	26.7545	6.1957	0.009
	Women	2805	12.8717	3.3837		785	27.3641	6.0431	
United States	Men	149,514	10.6281	2.0732	<0.001	62,949	27.1163	4.9311	<0.001
	Women	48,307	11.4174	2.1129		15,679	27.8919	4.5937	
South Africa	Men					3830	21.8187	3.3647	<0.001
	Women					585	23.1866	3.9734	
Other	Men Women	9469 1865	10.8642 11.4223	2.6745 2.7839	<0.001	12,864 2105	27.7271 28.1111	7.6246 7.5810	0.031


(Note: Due to the small sample size for 50-mile races, South Africa was combined with other countries; *p*-values are from comparisons of mean performances between sexes).

**Table 2 ijerph-16-02377-t002:** Regression analysis (mixed model) of ultra-marathons (50 and 100 miles). Estimates and standard errors (SEs) of fixed effects are reported. P-value ranges are marked with asterisks (see note). Smoothing terms, basis splines, are denoted with BS(x) t, where x = year, age; t = 1,2,3.

	50 miles Estimate (SE)	100 miles Estimate (SE)
Intercept	12.462 ^***^	23.658 ^***^
	(0.152)	(1.766)
**Year**		
BS (year) 1	−4.009 ^***^	1.399
	(0.236)	(2.416)
BS (year) 2	−1.218 ^***^	5.882 ^***^
	(0.126)	(1.609)
BS (year) 3	−0.313 ^*^	6.651^***^
	(0.148)	(1.750)
**Age**		
BS (age) 1	−2.427 ^***^	−8.204^***^
	(0.176)	(1.055)
BS (age) 2	−0.511^***^	3.674 ^***^
	(0.090)	(0.652)
BS (age) 3	4.039 ^***^	−0.637
	(0.189)	(1.523)
**Country (ref. United States)**		
Canada	−0.190 ^***^	−1.001^***^
	(0.040)	(0.156)
Great Britain	0.656 ^***^	−0.322 ^**^
	(0.028)	(0.109)
Other	−0.092 ^***^	0.893 ^***^
	(0.028)	(0.070)
South Africa		−3.938 ^***^
		(0.128)
**Sex (ref. Men)**		
Women (W)	0.740 ^***^	0.810 ^***^
	(0.017)	(0.075)
**Country*Sex**		
Canada*W	0.057	0.751^*^
	(0.074)	(0.311)
Great Britain*W	0.562 ^***^	−0.133
	(0.061)	(0.273)
Other countries*W	−0.191 ^**^	−0.339
	(0.067)	(0.180)
South Africa*W		0.129
		(0.331)
Observations	231,980	107,445

*Note:* * *p* < 0.05; ** *p* < 0.01; *** *p* < 0.001.

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
