# Peer review of "Women Reduce the Performance Difference to Men with Increasing Age in Ultra-Marathon Running"

_ijerph, 2019, doi:10.3390/ijerph16132377_

Round 1
Reviewer 1 Report
The study is very interesting and I think it is a very important advance to be able to know the differences between men and women in the ultra-marathon races.
The introduction is very good and considers all the necessary sections to theoretically support this study. The ethics committee number is missing.
The results of the study are adequate, using the T-Student for this, although regression is not the most appropriate method, an ANOVA could have been used. The figures of the results and the discussion should be considered to eliminate them since they generate some confusion.
The bibliographic references are mostly wrong, they should be completely revised, no doi appears, in some of them only volume but not pages appear
Author Response
Comments and Suggestions for Authors
The study is very interesting and I think it is a very important advance to be able to know the differences between men and women in the ultra-marathon races.
The introduction is very good and considers all the necessary sections to theoretically support this study. The ethics committee number is missing.
We added the number and it reads now ‘This study was approved by the Institutional Review Board of the Kanton St. Gallen, Switzerland, with a waiver of the requirement for informed consent of the participants, as the study involved the analysis of publicly available data (01-06-2010)’
The results of the study are adequate, using the T-Student for this, although regression is not the most appropriate method, an ANOVA could have been used. The figures of the results and the discussion should be considered to eliminate them since they generate some confusion.
We thank the expert reviewer for this comment. We understand that maybe ANOVA would have been easier to interpret, but the mixed regression analysis we performed, was indeed necessary to correct for clustered observations within runners, which participate more than once. ANOVA would have not accounted for this. So we think that the results and the figures, as shown, are important even if they could not be immediate to understand. To address this aspect, we added „It was acknowledged that analyses of variance (ANOVA) might have been easier to interpret; however, the mixed regression analysis was preferred, since it was necessary to correct for clustered observations within runners, who participate more than once. ANOVA would have not accounted for clustered observations“ in the methods. In addition, we revised the legends of figures for clarity.
The bibliographic references are mostly wrong, they should be completely revised, no doi appears, in some of them only volume but not pages appear
We agree with the expert reviewer and revised the references (e.g. abbreviations of journals were corrected and pages were added when they were missing). Checking the already published papers in IJERPH, we observed that doi is not reported in the references.
Reviewer 2 Report
The aim of this study was to analyze sex-specific performance in ultra-marathon running according to age and distance. The study is well founded. Highlight the quality of the literature review. The methodological process used is solid and well-founded. The results are clearly presented, complemented with tables and figures. The discussion of the results is too extensive. The conclusions are based on the results obtained.
I think it is necessary to summarize the discussion section as it is very extensive.
Congratulations to the authors for their work.
Author Response
Comments and Suggestions for Authors
The aim of this study was to analyze sex-specific performance in ultra-marathon running according to age and distance. The study is well founded. Highlight the quality of the literature review. The methodological process used is solid and well-founded. The results are clearly presented, complemented with tables and figures. The discussion of the results is too extensive. The conclusions are based on the results obtained.
I think it is necessary to summarize the discussion section as it is very extensive.
Congratulations to the authors for their work.
We agree with the expert reviewer and deleted the following parts (~550 words) from discussion:
a) From 4.1.: “Men were generally faster than women, statistically significantly so below 60 years of age in 100 miles races and below 75 years of age in 50 miles races. Regarding the gradient of the sex difference, our hypothesis of the sex difference in performance reducing with increasing age was confirmed. In both distances, the sex difference was most pronounced between 30 and 40 years of age. The decrease of sex difference in the older age groups compared to their younger counterparts might be partially explained by the variation of the number of finishers by sex and age group. The lower men-to-women ratio in the older than in the younger age groups indicated that elder women finishers might be more ‘competitive’ than men, which - in turn - might explain the relatively small sex difference.” and “For either distance, the range of sex differences between countries exceeds that within countries and is larger for 50 than for 100 miles races, the ranking of the countries being different for the two distances. A multitude of factors could be involved in bringing about this pattern, including, but not restricted to, climate, terrain, and sociocultural factors. As this information is not available, no further interpretation of these findings can be provided.”
b) From 4.5.: “Despite this impact of the US-specific phenomenon in shaping the performance model curve, the general trend visible in the data is a performance decline over the study period. One factor for this general decrease in running speed across calendar years could be the popularity of ultra-marathon races gradually increasing worldwide and that the races have increasingly attracted recreational (i.e. master) athletes [47]. As a consequence, the average performance would have gradually shifted to lower levels. It can be assumed that 100 miles races have always been less attractive to recreational athletes, as they require a more rigorous preparation than 50 miles races. An example for a more rigorous preparation with increasing race distances is provided by Rüst et al. [48] who compared training characteristics between marathoners and 100-km ultra-marathoners and found that ultra-marathoners completed significantly more hours and kilometers during their training. It is therefore likely that the influx of recreational athletes into 100 miles races has been more gradual then in for example 50 miles races, and this trend leads to a non-linearity of the performance trend curve.”
c) From 4.6: “it provided information on finishers, whereas data from non-finishing participants is not provided. This is unfortunate, as many of the above interpretations could have been further evaluated by such data. Potentially relevant covariates that could have been considered in the present analysis, but were not available, include, but are not limited to, ethnicity/race, weight, height, leg length, past injuries and concomitant diseases of the finishers, humidity and temperature or time of day of race onset. Of particular value would have been information on the length of running experience in general and of ultra-marathon running experience in particular, but this is not contained in the data set. Also, since the age of the finishers is only provided through birth year, a variability in the magnitude of up to almost one year is introduced which, however, is unlikely to have biased the results as it will have canceled out across participants. In spite of these limitations, the data set provides a wealth of relevant and consistent information, based on which the study novel results were obtained.”
Reviewer 3 Report
In my opinion this manuscript is beyond the scope of the journal ("broad
spectrum of important topics which are relevant to environmental health
sciences and public health protection"). It has no bearing either on
environmental nor on public health. I think the manuscript would be much
better placed in a sports science journal.
Author Response
Comments and Suggestions for Authors
In my opinion this manuscript is beyond the scope of the journal ("broad spectrum of important topics which are relevant to environmental health sciences and public health protection"). It has no bearing either on environmental nor on public health. I think the manuscript would be much better placed in a sports science journal.
Answer: We thank the expert reviewer for the opportunity to clarify the suitability of this paper for IJERPH. Actually, the paper has been submitted for consideration in the special issue Sports & Health (https://www.mdpi.com/journal/ijerph/special_issues/sports_health). In addition, we added about the relevance of the paper for health before the conclusions the part “In addition to performance, the abovementioned practical applications were also relevant from a health perspective. The role of exercise in the prevention and treatment of diseases (e.g. coronary artery disease, stroke, hypertension, diabetes, arthritis, osteoporosis, dyslipidemia, obesity, depression, cancer, and chronic obstructive pulmonary disease) has been well recognized [49]. The findings of the present study would aid physicians prescribing endurance exercise considering sex and age [50].”